# Single-cell transcriptome reveals the novel role of T-bet in suppressing the immature NK gene signature

Chao Yang[1,2†], Jason R Siebert[1,2†], Robert Burns[3], Yongwei Zheng[4], Ao Mei[1,2], Benedetta Bonacci[5], Demin Wang[4], Raul A Urrutia[6], Matthew J Riese[2,7,8], Sridhar Rao[9], Karen-Sue Carlson[8,10], Monica S Thakar[1,11], Subramaniam Malarkannan[1,2,8,11]*

[1]Laboratory of Molecular Immunology and Immunotherapy, Blood Research Institute, Blood Center of Wisconsin, Milwaukee, United States; [2]Department of Microbiology and Immunology, Medical College of Wisconsin, Milwaukee, United States; [3]Bioinfomatics Core, Blood Research Institute, Blood Center of Wisconsin, Milwaukee, United States; [4]Laboratory of B-Cell Lymphopoiesis, Blood Research Institute, Blood Center of Wisconsin, Milwaukee, United States; [5]Flow Cytometry Core, Blood Research Institute, Blood Center of Wisconsin, Milwaukee, United States; [6]Department of Surgery, Medical College of Wisconsin, Milwaukee, United States; [7]Laboratory of Lymphocyte Biology, Blood Research Institute, Blood Center of Wisconsin, Milwaukee, United States; [8]Department of Medicine, Medical College of Wisconsin, Milwaukee, United States; [9]Laboratory of Stem Cell Transcriptional Regulation, Blood Research Institute, Blood Center of Wisconsin, Milwaukee, United States; [10]Laboratory of Coagulation Biology, Blood Research Institute, Blood Center of Wisconsin, Milwaukee, United States; [11]Department of Pediatrics, Medical College of Wisconsin, Milwaukee, United States

*For correspondence:
subra.malar@bcw.edu

[†]These authors contributed equally to this work

Competing interests: The authors declare that no competing interests exist.

**Abstract** The transcriptional activation and repression during NK cell ontology are poorly understood. Here, using single-cell RNA-sequencing, we reveal a novel role for T-bet in suppressing the immature gene signature during murine NK cell development. Based on transcriptome, we identified five distinct NK cell clusters and define their relative developmental maturity in the bone marrow. Transcriptome-based machine-learning classifiers revealed that half of the mTORC2-deficient NK cells belongs to the least mature NK cluster. Mechanistically, loss of mTORC2 results in an increased expression of signature genes representing immature NK cells. Since mTORC2 regulates the expression of T-bet through $Akt^{S473}$-FoxO1 axis, we further characterized the T-bet-deficient NK cells and found an augmented immature transcriptomic signature. Moreover, deletion of *Foxo1* restores the expression of T-bet and corrects the abnormal expression of immature NK genes. Collectively, our study reveals a novel role for mTORC2-$Akt^{S473}$-FoxO1-T-bet axis in suppressing the transcriptional signature of immature NK cells.

## Introduction

NK cells are type one innate lymphoid cells known for their function in mediating both cytotoxicity and cytokine production in response to transformed or virally-infected cells (*Sun and Lanier, 2011*; *Vivier et al., 2008*). In the bone marrow (BM) of mice, the expression of IL-15/IL-2 receptor β chain (CD122) is the hallmark of commitment to the NK cell lineage from the common lymphoid progenitor (CLP) cells (*Rosmaraki et al., 2001*). Thus, the canonical definition of NK progenitor (NKP) is a

Lin$^-$CD122$^+$ cell without cell surface expression of NK-lineage marker NK1.1. However, the Lin$^-$CD122$^+$NK1.1$^-$ cells are still a heterogeneous population as approximately only one in ten cells can give rise to NK cells in vitro (*Vosshenrich and Di Santo, 2013*). Recently, the definition of NKPs has been refined to Lin$^-$Flt3$^-$CD27$^+$2B4$^+$CD127$^+$CD122$^+$NK1.1$^-$ cells which have a 50% chance to develop into NK cells in vitro (*Carotta et al., 2011*; *Fathman et al., 2011*). The other lineage cell type in the heterogenous Lin$^-$CD122$^+$NK1.1$^-$ population is yet to be fully-defined.

Following commitment, NK lineage cells go through step-wise developmental process to become functionally mature NK cells. The Lin$^-$CD122$^+$NK1.1$^+$CD27$^+$CD49b$^-$ population represents the least mature NK cells (*Di Santo, 2006*; *Kim et al., 2002*; *Williams et al., 2000*; *Arase et al., 2001*). Subsequent to the expression of CD49b, the developmental stages of murine NK cells further classified into three stages based on two cell surface markers CD27 and CD11b (*Kim et al., 2002*; *Hayakawa and Smyth, 2006*). The relatively immature CD27$^+$CD11b$^-$ (CD27 single positive, SP) NK cells develop into CD27$^-$CD11b$^+$ (CD11b single positive, SP) terminally mature with a transitional stage of CD27$^+$CD11b$^+$ (double positive, DP) cells (*Chiossone et al., 2009*). The terminal CD11b SP NK cells are also marked with expression of KLRG1 (*Huntington et al., 2007*).

The developmental process is controlled by temporal activation of stage-specific transcription factors. Ets-1 and PU.1 regulate the transition from CLPs to NKPs, while E4BP4 is critical in the induction of CD122 and NKP commitment (*Barton et al., 1998*; *Colucci et al., 2001*; *Gascoyne et al., 2009*; *Kamizono et al., 2009*). E4BP4 also induces the expression of Id2 and Eomes, both of which are critical in immature NK cells development and the transition to the later stages (*Male et al., 2014*; *Yokota et al., 1999*; *Boos et al., 2007*; *Delconte et al., 2016*; *Gordon et al., 2012*). T-bet and Zeb2 have been shown to be critical for terminal NK cell maturation (*Townsend et al., 2004*; *van Helden et al., 2015*). The signaling downstream of IL-15 receptors are essential for regulating the expression of these transcription factors. PDK1 is required for the induction of E4BP4 during early NK cell development (*Yang et al., 2015*). We have previously reported that mTORC1 is required for the expression of Eomes and the transition from CD27 SP to DP NK stage, while mTORC2 is required for the terminal CD11b SP NK cell maturation through mTORC2-Akt$^{S473}$-FoxO1 axis (*Yang et al., 2018*).

Although the cell surface markers have been useful in studying the development of NK cells, there are inherent limitations associated with them. The spectrum of NK cell developmental heterogeneity is unlikely to be fully defined by the current cell surface markers-defined stages. Based on the CD27/CD11b-defined stages, NK cells deficient in Id2, Eomes or Gata3 all fail to progress from CD27 SP to DP stages (*Delconte et al., 2016*; *Gordon et al., 2012*; *Ali et al., 2016*). Currently no well-defined model clearly explains developmental differences between these three NK models. In addition, the altered expression of CD27/CD11b in genetically-engineered mice does not necessarily result from developmental changes. The recent breakthrough in the quantification of transcriptome at a single-cell level has offered an unprecedented methodology in determining the cell identity and exploration of cellular heterogeneity. The definition of cell identities based on the expression of thousands of transcripts seems more reliable than the detection of limited number of cell surface markers. *Crinier et al., 2018* has explored the murine and human NK cells from spleen and blood using single-cell RNA-sequencing (scRNA-seq) technology and identified a novel population in the mouse spleen and revealed the conserved NK subsets in human and mice. However, the developmental heterogeneity of NK cells in the BM, the critical anatomic location of murine NK cell development, has not been explored.

In this study, we define the heterogeneity of murine CD3ε−CD122$^+$ cell from BM of wide type (WT) mice using scRNA-seq technology. The single-cell transcriptome analyses revealed three major cell types from the CD3ε−CD122$^+$ compartment: conventional NK cells, innate lymphoid cells 1 (ILC1), and cells with high transcripts contents of *Cd3d/e/g*. We define five distinct NK developmental subsets: an immature NK (iNK) cell cluster, three transitional NK (transNK) cell clusters, and a terminally mature NK (termNK) cell cluster. We explored the transcriptome-based cellular stage of Raptor- or Rictor-deficient NK cells. Unexpectedly, half of the Rictor-deficient NK cells are classified into the least mature iNK cluster which only comprise 25% of the WT NK cell in the BM. This is contradictory to cell surface markers-defined maturity with only the loss of terminally mature NK cells in *Rictor* conditional knockout (cKO) mice. As we previously proposed that mTORC2 regulates terminal NK cell maturation through promoting the expression of T-bet via Akt$^{S473}$-FoxO1 axis, we explored the maturation status of T-bet deficient NK cells using scRNA-seq. Strikingly, more than 65% of

T-bet-deficient NK cells are classified into the least mature iNK cluster and the expression of immature NK signature genes are highly up-regulated in the T-bet-deficient NK cells. Finally, deletion of *Foxo1* successfully rescued the developmental impairment of Rictor-deficient NK cells defined by both cell surface markers and developmental transcriptome markers. These findings revealed previously unappreciated role of mTORC2-Akt$^{S473}$-FoxO1-T-bet axis in suppressing the immature NK transcriptional signature during the development of NK cells.

## Results

### Single-cell transcriptome-based heterogeneity among CD3ε−CD122$^+$ cells

The BM is the anatomic location where most conventional murine NK cells develop. Thus, we decided to study the developmental heterogeneity of BM NK cells at single cell level using the 10X Genomics single cell gene expression system. To cover the broad NK cell developmental stages, we sorted the CD3ε−CD122$^+$ population from BM of the *Rptor*$^{fl/fl}$ *Ncr1*$^{Cre/WT}$, *Rictor*$^{fl/fl}$ *Ncr1*$^{Cre/WT}$, or *Tbx21*$^{−/−}$ mouse and the corresponding WT control mouse. The post-sorting purity ranged from 90% to 98% (*Figure 1—figure supplement 1A*). To validate the NK cell development phenotype specifically in these six mice used, we detected the expression of CD27 and CD11b via flow cytometry using the cells from BM and spleen of these six mice. Consistent with our previous observation (*Yang et al., 2018*), in the BM, most of the NK cells from the *Rptor*$^{fl/fl}$ *Ncr1*$^{Cre/WT}$ mouse were CD27 SP. The NK cells from the *Rictor*$^{fl/fl}$ *Ncr1*$^{Cre/WT}$ mouse were unable to fully progress to the CD11b SP stage (*Figure 1—figure supplement 1B*), and the T-bet-deficient mouse completely lost the CD11b SP NK compartment (*Figure 1—figure supplement 1B*; *Gordon et al., 2012*). The expression pattern of CD27 and CD11b on NK cells in the spleen also matched with previous reports (*Figure 1— figure supplement 1B*; *Gordon et al., 2012*; *Yang et al., 2018*). There was no difference in surface expression of CD27/CD11b among the three WT mice (*Figure 1—figure supplement 1B*).

After sequencing the libraries, the initial quality control (QC) analysis indicates successful library assembly, optimal sequencing and good cell viability. (*Figure 1—figure supplement 1C*). We started our analyses with a focus on exploring the heterogeneity of CD3ε−CD122$^+$ cells from WT mice using principal component analysis (PCA). To increase the clustering efficiency, cells from three WT mice were combined for analysis (*Andrews and Hemberg, 2018*). After initial quality control and clustering analysis, we filtered out the contaminating cells from other lineages, particularly those that did not express *Il2rb* (data not shown). The clustering analysis of the remaining pure CD3ε−CD122$^+$ cells revealed that the cells from *Tbx21*$^{+/+}$ mouse ordered from the Jackson Laboratory clustered separately from the *Rptor*$^{fl/fl}$ *Ncr1*$^{WT/WT}$ and *Rictor*$^{fl/fl}$ *Ncr1*$^{WT/WT}$ mice housed at the Medical College of Wisconsin animal facility (*Figure 1—figure supplement 1D and E*). These data revealed that different housing conditions could result in alteration in the transcriptome profile that leads to individual-specific phenotype in scRNA-seq analysis of NK cells.

Therefore, we scaled the sample variance to ensure the cells from different WT mice cluster together. At low clustering resolution, we found five distinct clusters of CD3ε−CD122$^+$ cells (*Figure 1A*). Through exploring the differential expressed genes (DEGs) of each cluster, we found that Cluster #1 were conventional NK cells with high expression of *Ncr1* and three subunits (*Klrb1a*, *Klrb1b*, and *Klrb1c*) of NK1.1 and comprised majority of the CD3ε−CD122$^+$ cells (*Figure 1A and B*). The expression of *Mki67* indicated that cells in Cluster #2 were cycling (*Figure 1B*). Cluster #3 contained ILC1 cells indicated by the high expression of *Tmem176a/b* and *Cxcr6* (*Figure 1B*; *Robinette et al., 2015*). Notably, *Ncr1* and *Klrb1a/b/c* were abundantly expressed in the ILC1 cluster that was higher than the NK cluster (*Figure 1B*). Cluster #4 was marked with high expression of *Cd3d/e/g* and low *Ncr1* and *Klrb1a/b/c* expression (*Figure 1B*). This population has been reported before in the scRNA-seq dataset of group 1 ILC in the lung and is potentially related to NK-T lineage (*Ferrari de Andrade et al., 2018*). We referred to it as *Cd3*$^{high}$ cluster. ILC1, *Cd3*$^{high}$ cells and NKP potentially make up the Lin−CD122$^+$NK1.1$^-$ cells. Cells in Cluster #5 were activated by inflammatory stimuli as made evident by the high expression of interferon-induced genes (*Figure 1B*). Clusters #2 and #5 cells did not express *Tmem176a/b*, *Cxcr6*, *Cd3d/e/g*, or other lineage-defining transcripts, and therefore, were likely to be conventional NK cells.

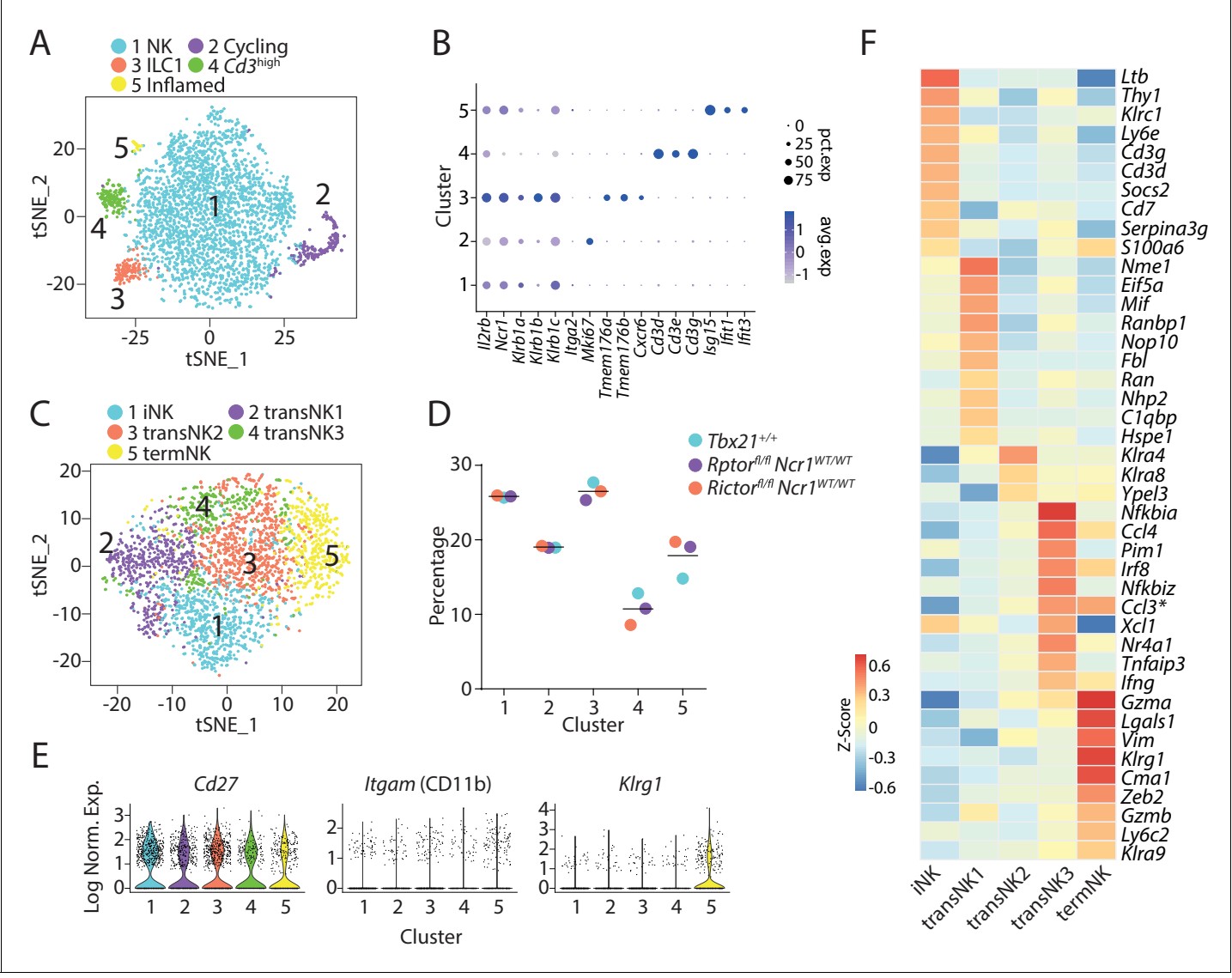

**Figure 1.** Transcriptome-based classification of CD3ε−CD122+ murine BM cells. (A) With low clustering resolution (0.2), the bulk CD3ε−CD122+ BM cells from three WT mice were clustered into five distinct populations demonstrated in the tSNE plot. (B) The expression of selected genes associated with the identity of each cluster in (A) were shown in a dot plot. The size of the dot indicates the percentage of cells expressing the gene within each cluster (pct.exp). The color of the dot indicates the average expression of the gene within each cluster (avg.exp). (C) Five distinct clusters identified through unbiased clustering analysis of the NK cluster in (A). (D) The composition of the five NK clusters in each of the WT mouse. (E) Violin plots demonstrate the expression of *Cd27*, *Itgam* (CD11b), and *Klrg1* in each NK cluster. The y-axis represents log-normalized expression value. (F) The average expression of the top 10 up-regulated DEGs (ranked by log fold-change) of each NK cluster were plotted using heatmap. The transNK2 cluster only contained three up-regulated DEGs with the 0.25 average log fold-change cutoff. * indicates genes that are DEGs of more than one cluster.

The online version of this article includes the following figure supplement(s) for figure 1:

**Figure supplement 1.** scRNA-seq analysis of CD3ε−CD122+ murine BM cells from three mutant mice and their corresponding WT mice.
**Figure supplement 2.** Unique features associated with tranNK1 cluster.

## Single-cell transcriptomic analysis of murine BM NK cells reveals five distinct clusters

Next, we removed Clusters #2 to #5 to focus on the analysis of the heterogeneity of the canonical NK cells. Unbiased clustering analysis revealed five distinct NK clusters as shown using a tSNE plot (*Figure 1C*). The three WT mice contributed relatively equal to each cluster, and each wild type had relatively similar proportions of each cluster (*Figure 1D*). Clusters #1 through #5 accounted for 25%,

20%, 25%, 10%, and 20% of the total NK cells, respectively (*Figure 1D*). Based on the expression of several NK cell maturation-defining markers, we found that Cluster #1 represented the most immature NK cells with high expression of *Cd27* and low expression of *Itgam* (CD11b) and Ly49s (*Klra1/3/4/7/8/9*) (*Figure 1E* and *Figure 1—figure supplement 1F*). The immature nature of Cluster #1 was further supported by the high transcriptional expression of *Ltb*, *Thy1*, *Cd3d/g*, and *Cd7* (*Figure 1F*), all of which have been shown to be highly expressed in the immature CD27 SP NK cells (*Chiossone et al., 2009*). In comparison, the low expression of *Cd27* and high expression of *Itgam* (CD11b) and *Klrg1* marked Cluster #5 as the terminally mature NK cells (*Figure 1E*). NK cells are known to acquire effector functions as they mature. The high expression of functional molecules including *Gzma* and *Gzmb* further demonstrated the terminal maturity of cells in Cluster #5 (*Figure 1F*). Based on this information, we deemed Cluster #1 and #5 as immature NK (iNK) cluster and terminally mature NK (termNK) cluster, respectively. We further defined Clusters #2, 3, and four as transitional NK stages (transNK1, 2, and 3, respectively).

Next, we sought to explore the identity of the three transNK clusters. TransNK1 represented a unique NK sub-population with high expression of genes encoding proteins involved in ribosomal biogenesis including ribonucleoproteins (*Nop10*, *Nhp2*, *Gar1*, *Npm1*, *Npm3*), RNA modification enzymes (*Mettl1*, *Ddx21*, *Fbl*), and GTPase related to nucleocytoplasmic transport (*Ran*, *Ranbp1*). We also found high expression of genes encoding the ribosomal subunits in this cluster (*Figure 1—figure supplement 2A*). Gene set enrichment analysis (GSEA) revealed significantly enrichment of transcription factors MYC and E2F in transNK1 cluster compared to all other cells, both of which are involved in cell growth and proliferation (*Figure 1—figure supplement 2B*; *Conacci-Sorrell et al., 2014*; *Chen et al., 2009*). We also found major metabolic pathways were enriched in this cluster including glycolysis, oxidative phosphorylation and fatty acid metabolism (*Figure 1—figure supplement 2C*). Fewer up-regulated transcripts were identified within the transNK2 cluster (*Figure 1F*). This cluster was featured with high expression of Ly49 family members especially *Klra4* (Ly49D) and *Klra8* (Ly49H) (*Figure 1F* and *Figure 1—figure supplement 1F*). As for the transNK3 cluster, we found multiple genes that belong to the category of immediate early genes (IEGs) were up-regulated (*Nr4a1*, *Nr4a2*, *Nr4a3*, *Dusp1*, *Junb*, *Nfkbia*, *Nfkbid*, *Nfkbiz*, *Egr1*, *Supplementary file 1*). This cluster is transcriptionally similar to the novel NK_3 cluster defined previously in the mouse spleen with up-regulation of genes including *Pim1*, *Gadd45b*, *Bhlhe40*, *Irf8*, *Ccl3*, *Ccl4* besides the IEGs (*Supplementary file 1*; *Crinier et al., 2018*).

To further elucidate the relative maturity of the three transitional NK clusters, we explored the transcriptomic progression along the maturation process. The Euclidean distance indicated transcriptional similarity between transNK1 and transNK2, both of which had higher transcriptional similarity to the iNK cluster (*Figure 2A*). In comparison, the transNK3 had higher transcriptional similarity to the termNK cluster (*Figure 2A*). As NK cells progress along the developmental stages, a gradual expression and loss of distinct set of genes are expected to occur temporally. Most of these genes are uniquely and abundantly expressed at the two developmental extremes: the immature and terminally mature stages. Therefore, we calculated the module score of all five clusters based on the up-regulated or down-regulated DEGs of the iNK and termNK clusters. When we ordered the five clusters as iNK→transNK1→transNK2→transNK3→termNK, the module score based on the up-regulated DEGs of the iNK cluster or the down-regulated DEGs of the termNK cluster gradually declined (*Figure 2B*). In contrast, the module score based on the up-regulated DEGs of the termNK or the down-regulated DEGs of the iNK cluster gradually increased (*Figure 2B*). This indicated a sequential developmental progression of these five clusters in the order listed above. To further validate this notion, we analyzed the bulk RNA-seq dataset of CD27 SP, DP and CD11b SP NK subsets published previously and defined the up-regulated DEGs from both CD27 SP and CD11b SP NK subsets as the target gene list (named as 'CD27/CD11b gene sets' here) to calculate the module score (*Figure 2C*; *Delconte et al., 2016*). The overall low module score yield indicated the low coverage of the transcriptome in the 10X-based scRNA-seq dataset. Nevertheless, we still found a trend in increasing expression of genes for terminally mature NK cells and decreasing expression of genes for immature NK cells validating the order of clusters (*Figure 2C*).

To further establish the developmental progression of these five clusters, we used the Monocle2 pseudotime analysis to simulate the maturation trajectory in an unbiased manner. Based on the transcriptional information, each cell from the five clusters were assigned to the pseudotime trajectory (*Figure 2D*). The pseudotime progression indicated the maturation order. As shown in *Figure 2E*,

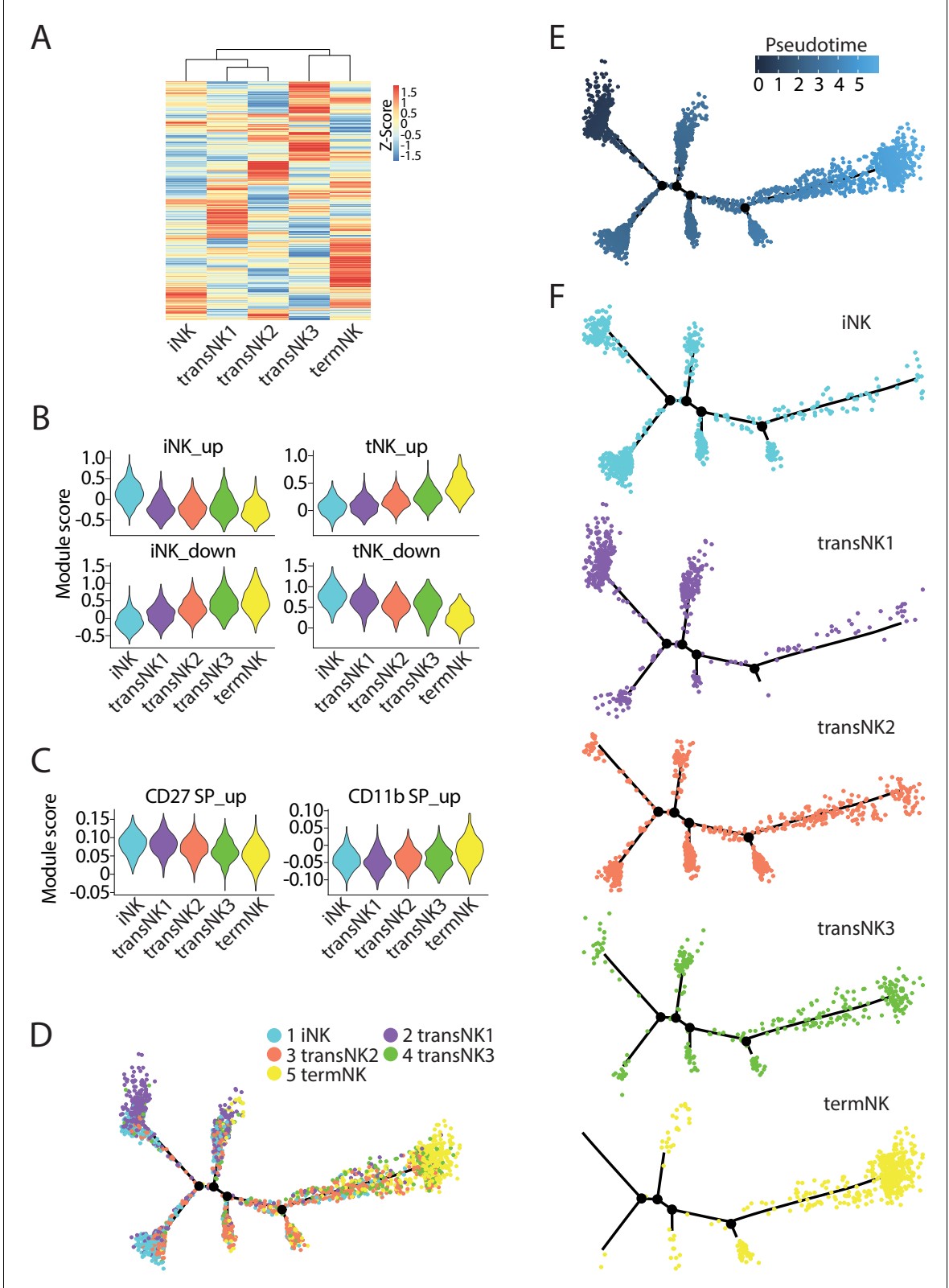

**Figure 2.** The relative maturity of the five distinct NK clusters. (**A**) The transcriptome similarity among the five NK clusters was evaluated by the hierarchical clustering analysis and visualized via heatmap. Each row represents one of the highly variable genes across all cells and each column represents the average expression of these genes within one cluster. (**B**) Module scores were calculated using up-regulated or down-regulated DEGs of iNK and termNK clusters and plotted via violin plots. (**C**) The up-regulated genes in the CD27 SP and CD11b SP subset were extracted from the CD27/

*Figure 2 continued on next page*

*Figure 2 continued*

CD11b bulk RNA-seq dataset. The expression level of these genes in the five NK clusters were evaluated via calculating module scores and plotted via violin plots. (D) Distribution of all five NK clusters along the pseudotime trajectory. (E) The relative maturity of the developmental trajectory displayed across pseudotime. (F) Distribution of each NK clusters along the pseudotime trajectory.

we picked the far-left side of the trajectory as the root point as we found cells from iNK and transNK1 clusters mostly resided here (*Figure 2F*). Based on this root point, the pseudotime dictated that the far-right side of the trajectory was the most mature population which was dominated by the termNK cluster (*Figure 2E and F*). The cells in transNK2 and transNK3 clusters spread across the trajectory (*Figure 2F*). In conclusion, based on the transcriptome signature associated with each cluster and the unbiased analysis of the maturation trajectory, we define the relative maturity of this five NK clusters as iNK→transNK1→transNK2→transNK3→termNK. It is important to point out that this relative maturity does not necessarily imply distinct developmental stages that all NK cells follow. As an example, the high expression of IEGs in transNK3 cluster implies an activated state. In this regard, transNK3 cluster may be a heterogenous population consists of NK cells from different developmental stages. Nevertheless, this relative maturity is instrumental in characterizing the NK cells from the knockout mice as shown below.

## Transcriptional alterations of raptor- or Rictor-deficient NK cells at single-cell level

The expression of cell surface markers has been conventionally used to define cellular identity. Although it is useful in many cases, there are limitations and potential drawbacks associated with this methodology especially in disordered conditions seen in patients or transgenic mice. Previously, we have characterized the development of Raptor- or Rictor-deficient NK cells that do not possess mTORC1 or mTORC2, respectively (*Yang et al., 2018*). We found differential maturation impairment with Raptor-deficient NK cells accumulating at the CD27 SP stage while Rictor-deficient NK cells are impaired in transition from DP to terminal CD11b SP stage. We profiled the BM $CD3^-CD122^+$ cells using scRNA-seq in these transgenic mice to validate whether transcriptome-based cell stage classification match with previous cell surface markers-defined maturity. We combined the mutant cells with their corresponding WT cells and conducted PCA-based clustering analysis. Strikingly, nearly all the Raptor-deficient $CD3^-CD122^+$ cells clustered distinctly from the WT cells (*Figure 3A*). As illustrated in the t-SNE plot, Cluster #1 through #5 were dominated by WT cells, while Cluster #6 and #7 were mostly Raptor-deficient cells (*Figure 3A* and *Figure 3—figure supplement 1A*). This is consistent with our previous demonstration of a large transcriptome alteration in the absence of Raptor (*Yang et al., 2018*). Cluster #8 is the only shared group with both Raptor-deficient and WT cells (*Figure 3A* and *Figure 3—figure supplement 1A*). The high expression of *Tmem176a/b* and *Cxcr6* revealed Cluster #8 were ILC1 while all the rest seven clusters were NK cells (*Figure 3—figure supplement 1B*). Comparatively, Cluster #1 had higher expression of *Cd3d/e/g*, however, this was not $Cd3^{high}$ cluster defined previously based on the high expression of *Ncr1*. Instead, cluster #1 were iNK cells and the higher transcripts level of CD3 chains has been reported before in immature NK cells (*Lanier et al., 1992*). We did not find cycling or inflamed cluster in this analysis presumably due to lower cell number from these two samples compared to the previous combined WT NK cell analysis (*Figure 3—figure supplement 1B*). As the expression of *iCre* driven by *Ncr1* presumably also occurred in the ILC1, this data indicated that mTORC1 is potentially dispensable for the development of ILC1.

Next, we examined the relationship between the five WT NK clusters and the two Raptor-deficient NK clusters. The DEGs revealed Cluster #1 and #5 represent the least and most mature NK clusters in the WT sample, respectively (*Supplementary file 2*). Module scores based on DEGs of iNK and termNK from above combined WT analysis supported this identity of Cluster #1 and #5 (*Figure 3B*). Consistent with their immature phenotype, Raptor-deficient NK cells from Cluster #6 and #7 expressed the termNK signature genes at low levels (*Figure 3B*). We calculated the module score based on the CD27/CD11b gene sets. The iNK and termNK features of Cluster #1 and #5 were consistent (*Figure 3—figure supplement 1C*). The Cluster #6 and #7 from the *Rptor* cKO mouse demonstrated similar level of CD27 SP stage signature gene expression as Cluster #1 and had lowest CD11b SP stage signature gene expression (*Figure 3—figure supplement 1C*). The

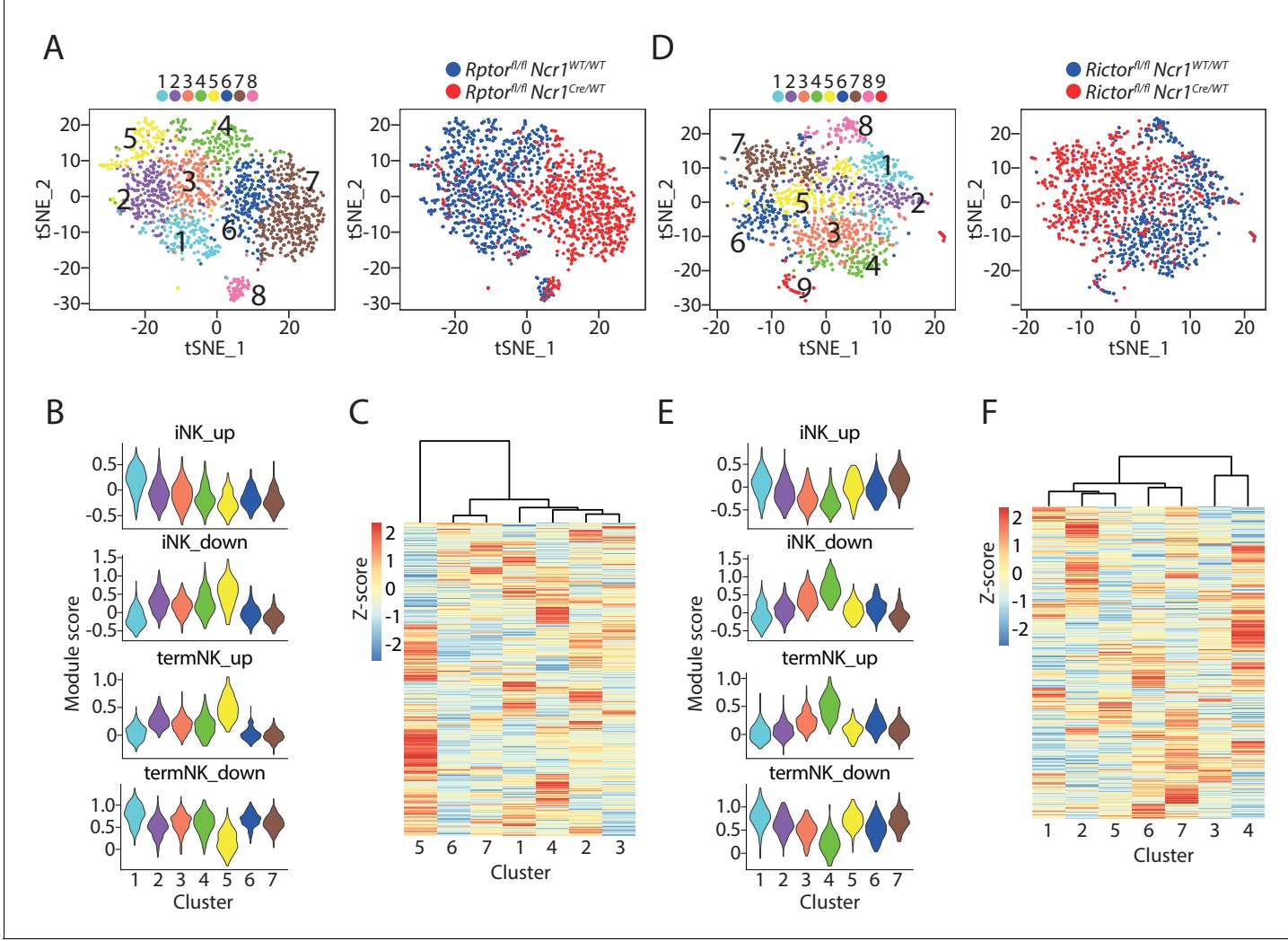

**Figure 3.** scRNA-seq analysis of Raptor- or Rictor-deficient NK cells. (A) Clustering analysis of CD3ε−CD122$^+$ cells from BM of *Raptor* cKO mouse and the littermate WT mouse. The clusters are displayed as tSNE plots on the left. The cell origin was labeled in the same tSNE plot on the right. (B) Module scores were calculated using up-regulated or down-regulated DEGs of iNK and termNK clusters and plotted in all the NK clusters formed by WT and Raptor-deficient cells. (C) The transcriptome similarity among the NK clusters formed by WT and Raptor-deficient cells was evaluated by the hierarchical clustering analysis and visualized via heatmap. (D–F) (D), (E) and (F) are same analysis using WT and Rictor-deficient NK cells as (A), (B), and (C), respectively.

The online version of this article includes the following figure supplement(s) for figure 3:

**Figure supplement 1.** Unbiased clustering analysis of Raptor- or Rictor-deficient NK cells.

Euclidean distance further emphasized the large transcriptome alteration of Raptor-deficient NK cells as Cluster #6 and #7 were distant from the WT clusters (*Figure 3C*).

Compared to Raptor-deficiency, Rictor-deficient NK cells have a more restricted transcriptomic alteration (*Yang et al., 2018*). Nevertheless, clustering analysis largely separated the WT NK cells with the Rictor-deficient NK cells (*Figure 3D* and *Figure 3—figure supplement 1D*). Within the nine identified clusters from the CD3ε−CD122$^+$ cells, Clusters #1 through #4 were mostly comprised with WT NK cells, while Clusters #5 through #7 were dominated by Rictor-deficient NK cells (*Figure 3D* and *Figure 3—figure supplement 1D*). We identified Cluster #8 and #9 as ILC1 and cycling cells indicated by the expression of *Tmem176a/b*, *Cxcr6* and Mki67, respectively (*Figure 3—figure supplement 1E*). The reduced contribution of *Rictor* cKO mice to the ILC1 cluster was potentially due to the impaired expression of T-bet which is critical to the ILC1 lineage (*Figure 3—figure supplement*

*1D*; *Klose et al., 2014*). The reduced cycling cells from the *Rictor* cKO mice was consistent with our previous report (*Yang et al., 2018*).

When we focused on the analysis of NK cells, the DEGs and module scores based on DEGs of iNK and termNK of combined WT analysis indicated that Cluster #1 and #4 represent the iNK and termNK in the *Rictor^fl/fl Ncr1^WT/WT* mouse, respectively, with Cluster #2 and #3 being the transitional stages (*Figure 3E* and *Supplementary file 3*). Interestingly, all three Rictor-deficient NK cells-dominated clusters (#5, 6, 7) had high expression of up-regulated genes in the iNK cluster to the level similar to the least mature Cluster #1 of WT NK cells (*Figure 3E*, iNK_up). The module scores based on CD27/CD11b gene sets demonstrated similar phenomenon, though less striking (*Figure 3—figure supplement 1F*, CD27 SP_up). The relative expression of signature genes associated with terminally mature NK cells in Rictor-deficient NK clusters (#5, 6, 7) was similar to the transitional NK clusters (#2, 3) from WT mouse (*Figure 3E* and *Figure 3—figure supplement 1F*). Strikingly, the hierarchical clustering analysis indicated that nearly all the Rictor-deficient NK cell-dominated clusters (#5, 6, 7) had more transcriptome similarity to the most immature cluster #1 and the less mature transitional cluster #2 of WT mouse (*Figure 3F*). The fact that cluster #1 and #2 only accounted for less than 50% of the WT NK cells emphasized that the Rictor-deficient NK cells were less mature than previously cell surface CD27/CD11b-defined maturity (*Figure 1—figure supplement 1B*).

## Transcriptome-defined maturity of raptor- or Rictor-deficient NK cells

As both Raptor- and Rictor-deficient NK cells clustered separate from their corresponding WT NK cells, we decided to apply machine-learning-based algorithm to classify the developmental stages of these genetically modified NK cells based on the WT controls. In total, we utilized five different machine-learning algorithms including generalized linear classifier (GL), gradient boosting classifier (GB), extreme gradient boosting classifier (XGB), random forest classifier (RF), and deep learning classifier (DL, neural networks). To train the classifiers, we randomly sampled 80% of the cells from the combined WT analysis. We tested the prediction accuracy of each classifier using the remaining 20% of the cells which were not included in the initial training process. As summarized in the *Figure 4—figure supplement 1A*, the overall accuracy of all classifiers ranged from 60% to 70% with deep learning (DL) classifier being the most accurate with the lowest error rate. Importantly, all five NK clusters were identified by the classifiers except for the RF classifier which failed to detect the transNK3 subset (*Figure 4A*). The overall composition of the five NK clusters assigned by the machine-learning classifiers were similar to the original identity defined in the PCA analysis (*Figure 4A*).

When we applied the classifier algorithms to the *Rptor* cKO samples, we barely detected transNK3 and termNK clusters (*Figure 4B and C* and *Figure 4—figure supplement 1B*). The most dominant cluster of the *Rptor^fl/fl Ncr1^Cre/WT* mouse was transNK2 accounting for more than 50% of the total Raptor-deficient NK cells (*Figure 4C*). The classification of Rictor-deficient NK cells resulted in 50% of all cells fell into the category of iNK cluster (*Figure 4D and E* and *Figure 4—figure supplement 1C*). This is a drastic contradiction to the cell surface markers-defined maturity which revealed only terminal maturation defects of Rictor-deficient NK cells (*Figure 1—figure supplement 1B*). This result was consistent with the high expression of iNK signature genes in *Rictor* cKO NK cells and short Euclidean distance between the WT immature NK compartments and all the Rictor-deficient NK cells (*Figure 3E and F*). Unlike the *Rptor* cKO sample, we could still detect transNK3 and termNK clusters in *Rictor* cKO NK cells with a reduction in percentage of termNK clusters compared to the littermate WT mouse consistent with reduced CD11b SP cells in *Rictor* cKO mice (*Figure 4A and E*). In summary, through machine-learning based classification, we found new insights into the maturity of Rictor-deficient NK cells that were previously masked by the cell surface markers-based definition.

## T-bet-deficient NK cells had a transcriptional profile similar to the iNK cluster

In our previous report, we proposed that mTORC2 regulates the terminal maturation of NK cells through Akt^S473-FoxO1-T-bet axis (*Yang et al., 2018*). As the impaired expression of T-bet in Rictor-deficient NK cells is potentially responsible for the maturation defects, we reasoned whether a large proportion of the T-bet-deficient NK cells are also transcriptionally similar to immature NK cells in

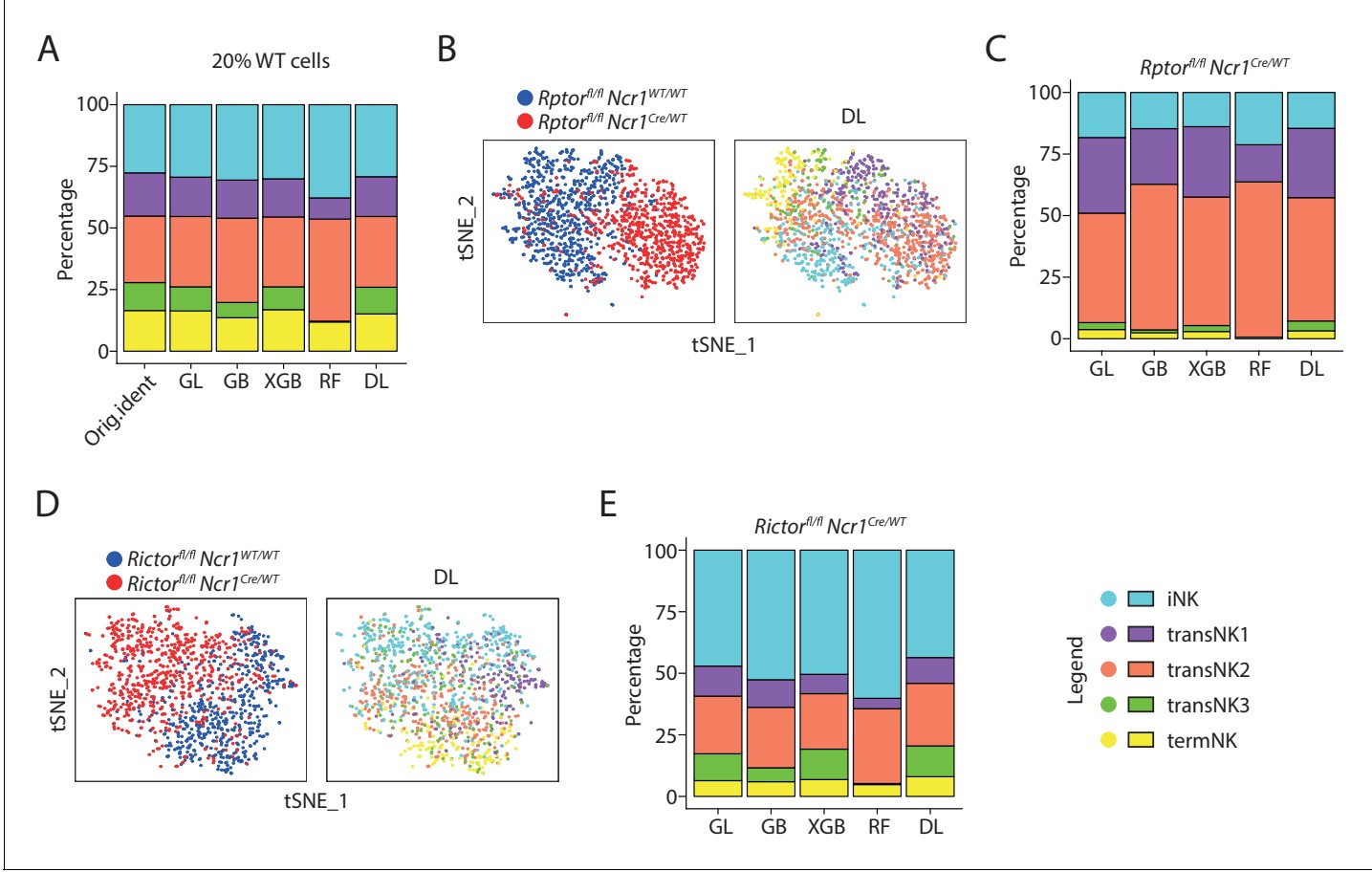

**Figure 4.** The identity of the Raptor- or Rictor-deficient NK cells defined by machine-learning classifiers. (A) The original composition of the five NK clusters in the 20% WT testing cells (referred as 'Orig.ident') along with the composition determined by the machine-learning classifiers were plotted in the bar graph. GL, GB, XGBoost, RF, and DL represents generalized linear classifier, gradient boosting classifier, extreme gradient boosting classifier, random forest classifier, and deep learning classifier, respectively. (B) The identity of each WT and Raptor-deficient NK cells were assigned by the deep learning classifier as shown in the tSNE plots. (C) The composition of the five NK clusters in Raptor-deficient NK cells determined by the machine-learning classifiers. (D, E) (D) and (E) are same analysis using WT and Rictor-deficient NK cells as (B) and (C), respectively.

The online version of this article includes the following figure supplement(s) for figure 4:

**Figure supplement 1.** The accuracy of the five machine-learning classifiers.

WT, a phenomenon seen in *Rictor* cKO NK cells. To test this, we conducted the scRNA-seq experiment using $CD3\epsilon^-CD122^+$ cells from BM of WT and $Tbx21^{-/-}$ mice purchased from the Jackson Laboratory. The PCA-based clustering analysis of $CD3\epsilon^-CD122^+$ cells from WT and *Tbx21* KO mice resulted in 12 distinct clusters with nearly complete separation between WT and *Tbx21* KO cells (*Figure 5—figure supplement 1A and B*). Based on the expression of key markers (*Figure 5—figure supplement 1C*), we found that Clusters #1–7 comprised of NK cells. Clusters #8–10 were ILC1, $Cd3^{high}$, and inflamed clusters, respectively. Both Cluster #11 and #12 were cycling cells with high expression of *Mki67* (Ki-67) (*Figure 5—figure supplement 1C*). The G2M.Score (module score of genes associated with G2M phase of cell cycle) and S.Score (module score of genes associated with S phase of cell cycle) indicated that cells in the Cluster #11 were mostly in the S phase while cells in the Cluster #12 were mostly in the G2M phase (data not shown). Illustrated by the distribution, there was almost no ILC1 (#8) and $Cd3^{high}$ (#9) cluster cells in the $Tbx21^{-/-}$ sample (*Figure 5—figure supplement 1B*). This was consistent with T-bet being the master transcription factor of ILC1 lineage (*Klose et al., 2014*), and the $Cd3^{high}$ cluster being a lineage related to NK-T cells which is nearly absent in $Tbx21^{-/-}$ mice (*Townsend et al., 2004*). The dominance of inflamed Cluster (#10) by

T-bet-deficient NK cells indicated perturbation of inflammatory response resulted from global deletion of *Tbx21* (*Figure 5—figure supplement 1B*).

Next, we focused on the analysis of Clusters #1 through #7 comprised of NK cells. DEGs and module scores indicated that Cluster #1 and #4 were the least and most mature NK cluster of WT mouse, respectively (*Figure 5A* and *Supplementary file 4*). All three T-bet-deficient NK cells-dominated clusters (#5, 6, 7) presented similar module score pattern as the immature cluster #1 in all up- or down-regulated DEGs of iNK or termNK cluster (*Figure 5A*). This pattern was also present in the module scores calculated based on the CD27/CD11b gene set (*Figure 5—figure supplement 1D*). The transcriptional similarity between the least immature WT NK cells (Cluster #1) and the bulk T-bet-deficient NK cells (Clusters #5, #6, #7) were further manifested by the hierarchical clustering analysis (*Figure 5B*). The short Euclidean distance among these clusters was even more striking when compared to the Rictor-deficient NK sample (*Figure 3F*). When we applied the machine-learning-based classifiers, more than 60% of the T-bet-deficient NK cells were categorized into the iNK cluster and there were only few cells in the transNK3 and termNK clusters (*Figure 5C and D* and *Figure 5—figure supplement 1E*). Collectively, these data revealed that T-bet-deficient NK cells were less mature than previously defined by cell surface markers. Direct comparison of the bulk WT and *Tbx21*$^{-/-}$ NK cells in the scRNA-seq dataset revealed that T-bet deficiency resulted in up-regulation of immature NK signature genes and down-regulation of terminally mature NK genes (*Figure 5E*). Within the 14 up-regulated genes (average logFC $\geq$0.25) in the iNK cluster from three WT combined analysis, we found 10 of those genes with significantly increased expression in T-bet-deficient NK cells compared to WT NK cells (*Figure 5E*). Within the 56 termNK cluster signature genes (average logFC $\geq$0.25), we also observed decreased expression of 18 genes in T-bet-deficient NK cells compared to WT NK cells (*Figure 5E*). These results imply that the large proportion of T-bet-deficient NK cells are halted at the transcriptionally immature NK cell stage.

The technical limitation of 10X-based scRNA-seq with capture of the most abundant transcripts only covered partial iNK signature transcripts. The extent to which T-bet suppresses the expression of immature NK signature genes is still unknown. To achieve a fair comparison and in-depth coverage of the transcriptome, we conducted bulk RNA-seq analysis using only the CD27$^+$ NK cells from BM of both *Tbx21* WT and KO mice. We first validated the expression of those 28 genes identified in the scRNA-seq dataset (*Figure 5E*). As demonstrated in the heatmap, indeed nearly all the 10 iNK signature genes were significantly up-regulated and the 18 termNK signature genes were significantly down-regulated in the T-bet-deficient NK cells comparing to the WT cells (*Figure 5F*). To investigate whether this up-regulation of immature NK signature genes is intrinsic to the loss of T-bet, we conducted mixed bone marrow chimera experiment and performed bulk RNA-seq analysis using CD27$^+$ NK cells from BM. Consistent with previous report (*Townsend et al., 2004*), we found absence of CD11b SP NK cells and significantly reduced KLRG1 expression on NK cells developed from T-bet KO donors (*Figure 5—figure supplement 2A and B*). RNAseq analysis indicated up-regulation of immature NK genes in the absent of T-bet, emphasizing cell-intrinsic defects (*Figure 5—figure supplement 2C*). Through analyzing the bulk RNA-seq dataset from Rictor-deficient CD27$^+$ NK cells we published previously (*Yang et al., 2018*), we found a similar pattern when we evaluated the expression of these 28 genes, consistent with the immaturity of the Rictor-deficient NK cells (*Figure 5—figure supplement 2D*). Next, to obtain a larger immature NK transcriptional signature, we analyzed the CD27/CD11b bulk RNA-seq dataset and found genes that have highest expression in the CD27 SP NK subset with more than one-fold increased expression comparing to both DP and CD11b SP NK cells. In total, we found 796 genes representing the transcriptional signature of the relatively immature CD27 SP NK subset. GSEA revealed that these 796 immature NK signature genes were significantly enriched in the T-bet-deficient NK cells comparing to the WT counterparts, emphasizing that T-bet is required to suppress the expression of genes associated with immature NK cells to a large extent (*Figure 5G*). In summary, these data revealed the previously unappreciated role of T-bet in suppressing the immature NK transcriptional signature during the development of NK cells.

## T-bet is unlikely to directly bind and suppress the expression of the bulk immature NK genes

With the novel finding that T-bet is required for the suppression of immature NK transcriptional signature, we sought to explore the potential mechanisms for the induction of immature NK genes in

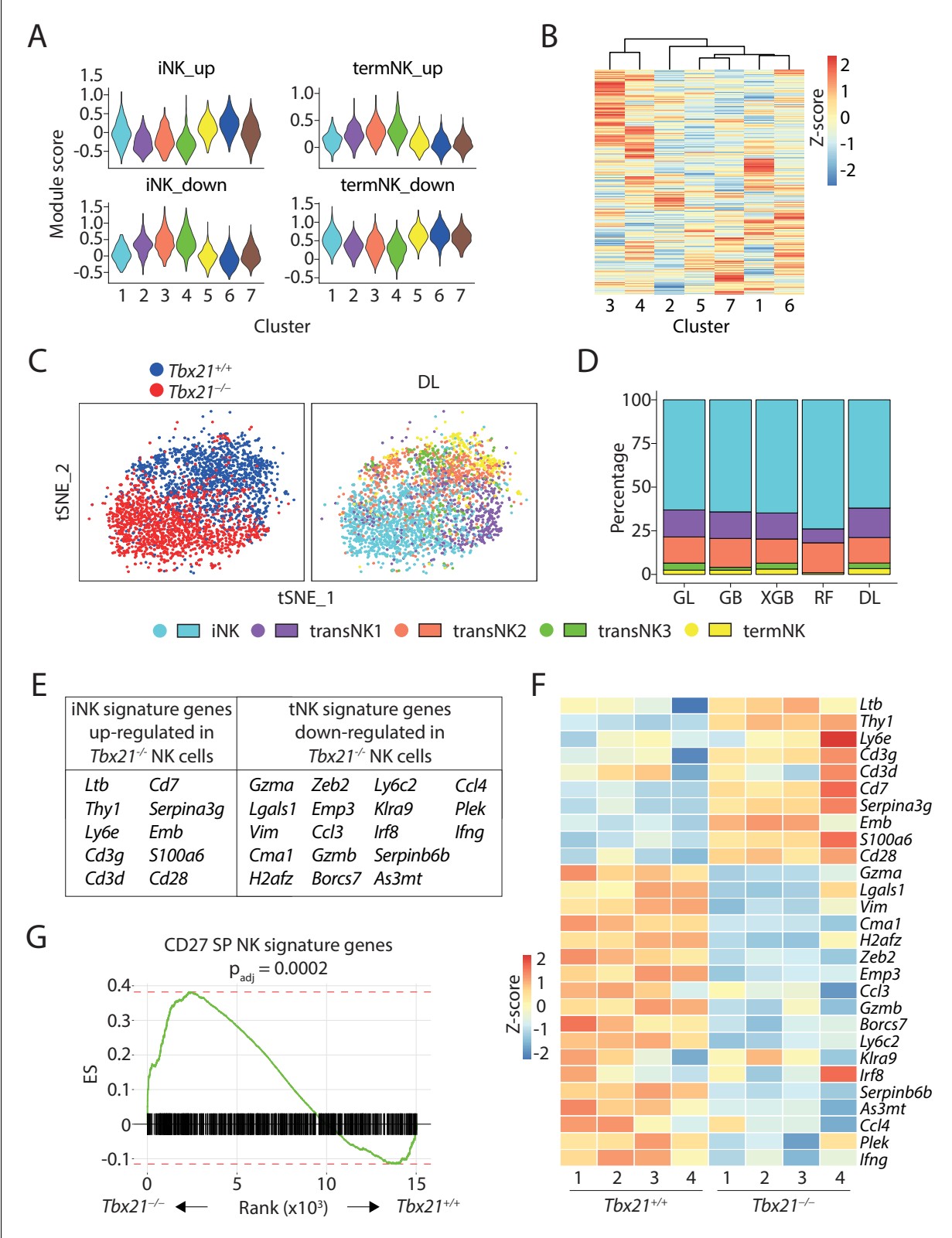

**Figure 5.** scRNA-seq analysis of T-bet-deficient NK cells. (**A**) Module scores were calculated using up-regulated or down-regulated DEGs of iNK and termNK clusters and plotted in all the NK clusters formed by WT and T-bet-deficient cells. (**B**) The transcriptome similarity among the NK clusters formed by WT and T-bet-deficient cells was evaluated by the hierarchical clustering analysis and visualized via heatmap. (**C**) The identity of each WT and T-bet-deficient NK cells were assigned by the deep learning (DL) classifier as shown in the tSNE plot. (**D**) The composition of the five NK clusters in

*Figure 5 continued on next page*

Figure 5 continued

T-bet-deficient NK cells determined by the machine-learning classifiers. GL, GB, XGBoost, RF, and DL represents generalized linear classifier, gradient boosting classifier, extreme gradient boosting classifier, random forest classifier, and deep learning classifier, respectively. (E) With a minimum of 0.25 average log fold-change threshold, the iNK signature genes that were significantly up-regulated and the termNK signature genes that were significantly down-regulated in T-bet-deficient NK cells compared to WT NK cells were listed in the table. (F) The expression level of genes listed in (E) was further validated in the bulk RNA-seq analysis of the CD27$^+$ WT and T-bet-deficient NK cells and shown in the heatmap. (G) The enrichment of CD27 SP NK subset signature genes in the T-bet-deficient NK cells compared to the WT NK cells.

The online version of this article includes the following figure supplement(s) for figure 5:

**Figure supplement 1.** Unbiased clustering analysis of CD3ε−CD122$^+$ cells from BM of the *Tbx21* KO mouse.
**Figure supplement 2.** Up-regulation of immature NK genes is intrinsic due to the loss of T-bet.

T-bet-deficient NK cells. One potential explanation is that T-bet, as a transcription factor, directly binds to the regulatory elements associated with immature NK genes and actively suppresses the expression of these genes. we reanalyzed the recently published T-bet ChIP-seq dataset generated using splenic NK cells and assessed whether T-bet directly targets the immature NK genes (*Shih et al., 2016*). With q-value set at $1 \times 10^{-5}$ and focus on the protein-coding genes, we found a list of 654 T-bet-binding genes. Next, we used the GSEA to determine whether the T-bet-binding genes were enriched in the signature genes associated with either iNK cluster or CD27 SP NK sub-set. We found a significant depletion of T-bet-binding genes in the iNK cluster compared to all other cells, indicating that T-bet does not directly bind to the majority of these iNK signature genes (*Figure 6A*, left). On the contrary, we found the T-bet-binding genes were significantly enriched in the termNK cluster comparing to all other cells (*Figure 6A*, right). Similarly, when comparing the CD27 SP with the CD11b SP NK subset in the bulk RNA-seq dataset, we found significant enrichment of the T-bet ChIP-seq peaks in the CD11b SP NK cells compared to the CD27 SP compartment (*Figure 6B*). These data were consistent with T-bet being the master transcription factor in driving terminal maturation of NK cells. Nevertheless, we found several immature NK signature genes that are up-regulated in T-bet-deficient NK cells with direct T-bet binding of the regulatory regions (*Figure 6C*). The expression of these genes might be subject to direct suppression from T-bet. In conclusion, T-bet is unlikely to directly bind and suppress the bulk of the immature NK genes.

Our second hypothesis was that T-bet regulates the expression and/or activity of other transcription factors which in turn regulate the expression of immature NK signature genes. We examined the expression of transcription factors that are known to play a role in the development of NK cells and found majority of them have an expression profile in either T-bet- or Rictor-deficient NK cells comparable to the WT (*Figure 6—figure supplement 1A and B*; *Hesslein and Lanier, 2011*). Next, we explored the GSEA and looked for enriched or depleted transcription factors in both T-bet- and Rictor-deficient NK cells. Interestingly, we found that transcription factor FoxO1 was enriched in the T-bet-deficient NK cells comparing to WT controls (*Figure 6D*), a phenomenon that we have previously established in the Rictor-deficient NK cells (*Yang et al., 2018*). We also found the transcripts level of *Foxo1* was significantly higher in the T-bet-deficient NK cells indicating that T-bet may suppress the expression of *Foxo1* (*Figure 6E*). The direct binding of T-bet in the intron region and the regulatory region up-stream of transcription starting site of *Foxo1* locus further supported this possibility (*Figure 6F*). These data prompted us to hypothesize that Foxo1, which has the highest expression in the CD27 SP subset (*Deng et al., 2015*), is essential in driving the expression of immature NK signature genes.

## Deletion of *Foxo1* rescues the maturation of Rictor-deficient NK cells

Previously we proposed that hyperactive FoxO1 suppresses the expression of T-bet in Rictor-deficient NK cells leading to the terminal maturation defects (*Yang et al., 2018*). Based on the data we presented above, we hypothesized that FoxO1 promotes the expression of immature NK genes in both T-bet- and Rictor-deficient NK cells. Therefore, we asked the question whether deletion of *Foxo1* in Rictor-deficient NK cells could not only rescue the impaired expression of T-bet and the terminal maturation defect, but also correct the abnormal induction of the immature NK transcriptional signature. To address this question, we bred the *Rictor$^{fl/fl}$Ncr1$^{Cre/WT}$* mice with *Foxo1$^{fl/fl}$Ncr1$^{WT/WT}$* mice. To ensure efficient generation of experimental mice with littermate control, we used *Rictor$^{fl/+}$Foxo1$^{fl/+}$Ncr1$^{Cre/WT}$* mice as the WT control mice since both Rictor and FoxO1 are

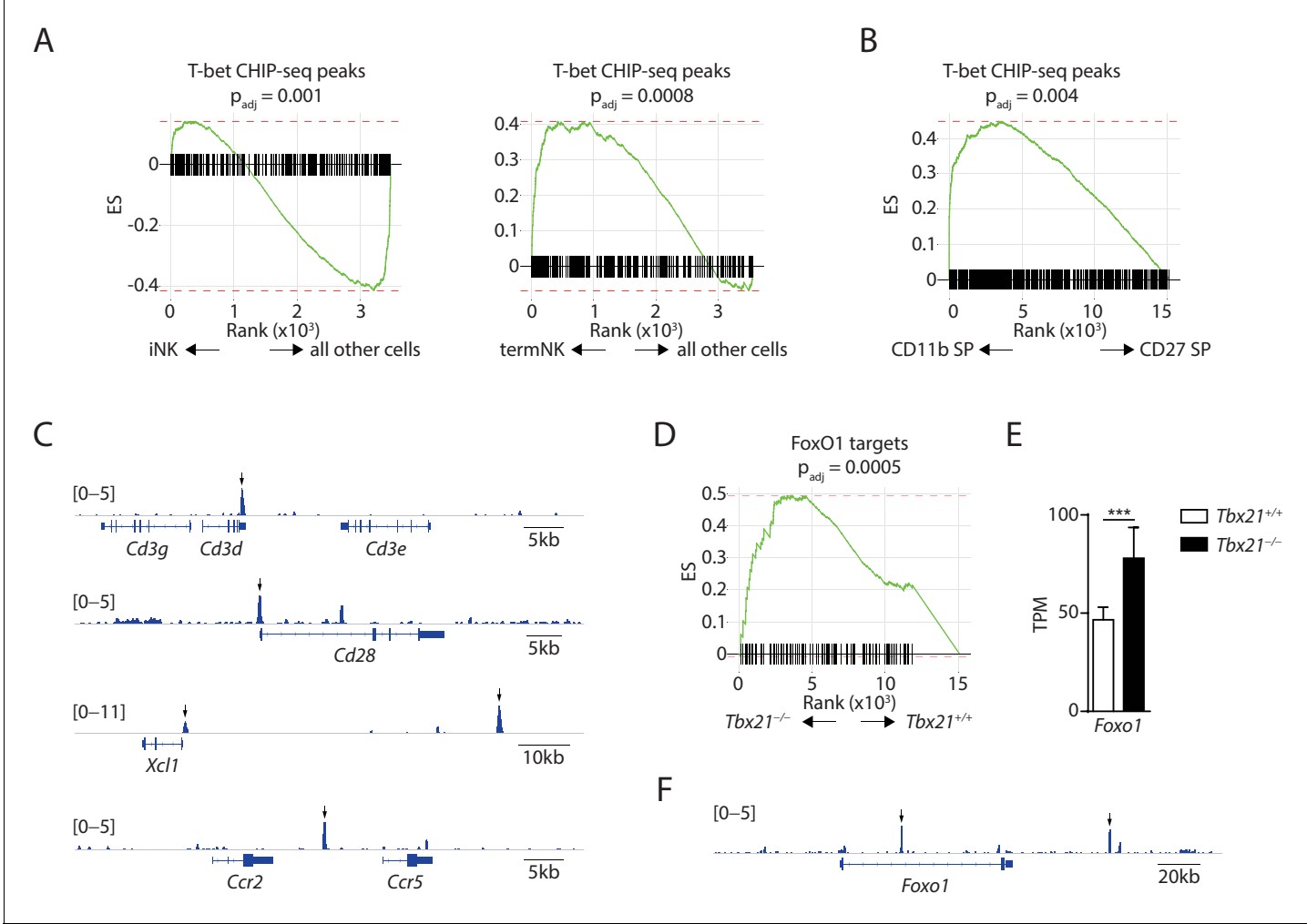

**Figure 6.** Exploration of the mechanisms underline the suppressive role of T-bet in regulating the immature NK transcriptional signature. (A) Depletion of the T-bet ChIP-seq peaks in the iNK cluster compared to all other cells (left) and enrichment of the T-bet ChIP-seq peaks in the termNK cluster compared to all other cells (right). (B) The T-bet ChIP-seq peaks were significantly enriched in the CD11b SP NK subset compared to the CD27 SP NK subset. (C) Examples of immature NK genes up-regulated in T-bet-deficient NK cells with significant binding of T-bet in the regulatory elements. Arrows point the significantly enriched peaks. (D) Enrichment of transcription factor FoxO1 in the T-bet-deficient NK cells compared to the WT NK cells. (E) The transcripts level of *Foxo1* in the CD27[+] WT and T-bet-deficient NK cells. TPM stands for transcripts per million. (F) Significant enrichment of T-bet ChIP-seq peaks at the regulatory elements associated with the *Foxo1* locus. Arrows point the significantly enriched peaks.

The online version of this article includes the following figure supplement(s) for figure 6:

**Figure supplement 1.** T-bet- or Rictor-deficiency does not result in large alteration in the expression of known transcription factors critical for the development of NK cells.

haplo-sufficient to the development of NK cells (data not shown). Due to the mixed background of these mice, we had large variation among each genotype most pronounced in terms of the number of NK cells in these mice (*Figure 7—figure supplement 1A and B*). Nevertheless, consistent with previous report (*Yang et al., 2018*), we found a decrease in both percentage and absolute number of NK cells in the spleen of *Rictor* cKO mice compared to the WT control (*Figure 7—figure supplement 1A and B*). Importantly, deletion of *Foxo1* in *Rictor* cKO mice rescued the number of NK cells in the spleen to the level compatible to the WT control (*Figure 7—figure supplement 1A and B*). Specific to the maturation of NK cells, *Rictor* cKO mice had a smaller terminally-mature population compared to the WT control indicated by either the percentage of CD11b SP subset or the KLRG1[+] NK cells (*Figure 7A and B*, *Figure 7—figure supplement 1C and D*), although the difference is less

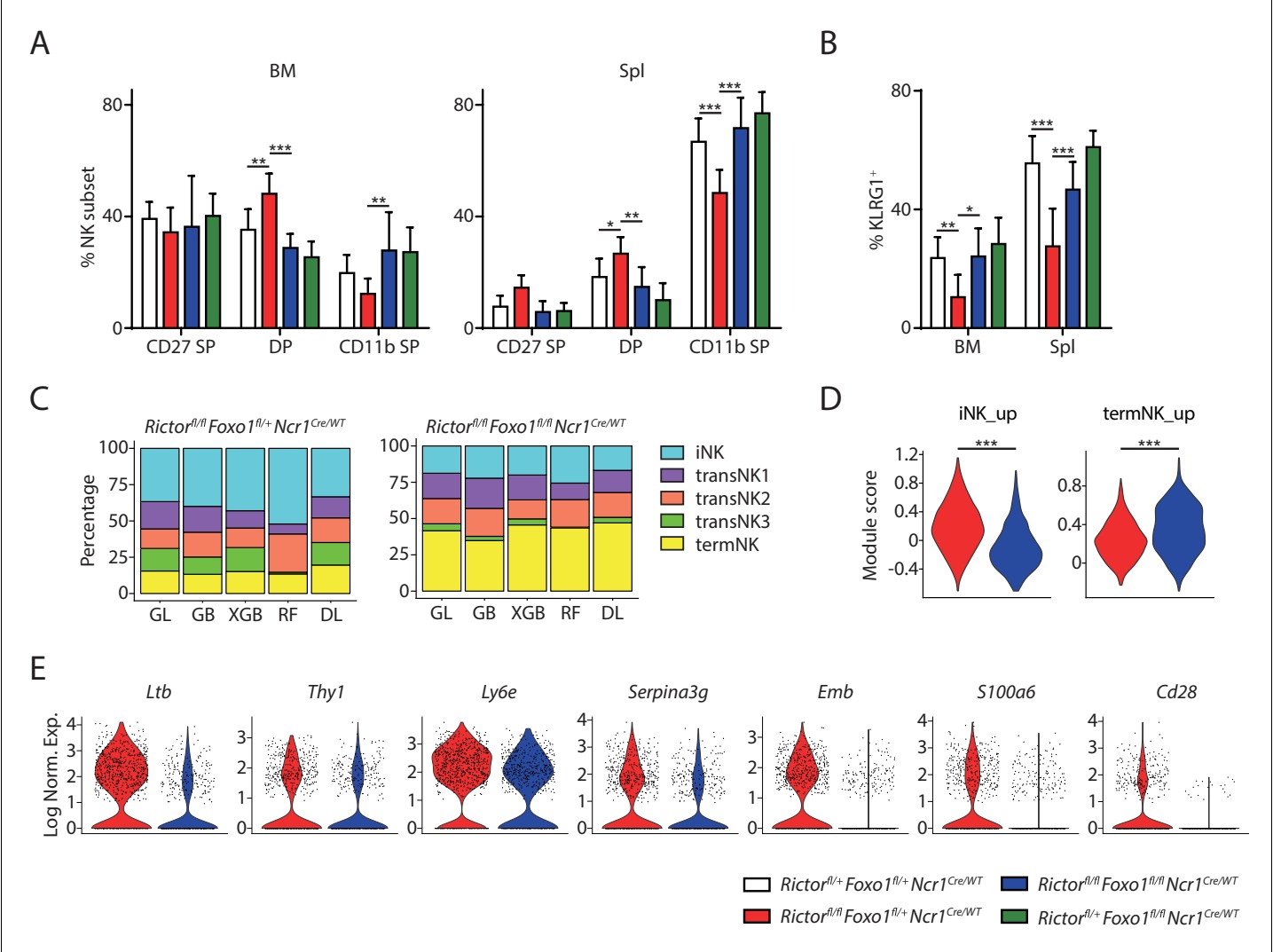

**Figure 7.** Deletion of *Foxo1* rescues the developmental defects in Rictor-deficient NK cells. (**A**) Quantification of each NK subsets defined by the expression of CD27 and CD11b in BM and spleen of the four group mice (gated on CD3ε−NCR1$^+$). n ≥ 6, pooled from six independent experiments. (**B**) Quantification of the percentage KLRG1$^+$ NK cells in the four group mice (gated on CD3ε−NCR1$^+$). n ≥ 6, pooled from six independent experiments. (**C**) The composition of the five NK clusters in the littermate *Rictor$^{fl/fl}$Foxo1$^{fl/+}$Ncr1$^{Cre/WT}$* and *Rictor$^{fl/fl}$Foxo1$^{fl/fl}$Ncr1$^{Cre/WT}$* mice determined by the machine-learning classifiers. GL, GB, XGBoost, RF, and DL represents Generalized Linear classifier, Gradient Boosting classifier, extreme gradient boosting classifier, Random Forest classifier, and Deep Learning classifier, respectively. (**D**) The expression of iNK and termNK signature genes in NK cells from *Rictor$^{fl/fl}$Foxo1$^{fl/+}$Ncr1$^{Cre/WT}$* and *Rictor$^{fl/fl}$Foxo1$^{fl/fl}$Ncr1$^{Cre/WT}$* mice were evaluated by module scores and plotted as violin plots. (**E**) The expression of several iNK signature genes in NK cells from *Rictor$^{fl/fl}$Foxo1$^{fl/+}$Ncr1$^{Cre/WT}$* and *Rictor$^{fl/fl}$Foxo1$^{fl/fl}$Ncr1$^{Cre/WT}$* mice were demonstrated via violin plots. Statistical significance was calculated using Two-way ANOVA (**A** and **B**) or unpaired Student t.test (**D**). *p<005; **p<0.01; ***p<0.001. The online version of this article includes the following figure supplement(s) for figure 7:

**Figure supplement 1.** Deletion of *Foxo1* rescues the NK cellularity and maturation defects in *Rictor* cKO mice.

pronounced compared to the previous report likely due to the fact that the mixed background mice had generally more mature NK cells than the pure B6 background mice at the age of eight weeks (*Figure 7—figure supplement 1C and D* and *Figure 1—figure supplement 1B*). Notably, deletion of *Foxo1* in Rictor-deficient NK cells completely rescued the percentage of CD11b SP subset or the KLRG1$^+$ NK cells to the level compatible to the WT control (*Figure 7A and B*, *Figure 7—figure supplement 1C and D*). These data indicated that FoxO1 is the transcription factor suppressing the terminal maturation program in Rictor-deficient NK cells. To further explore whether the impaired expression of T-bet is the downstream of hyperactive FoxO1 in Rictor-deficient NK cells, we

evaluated the expression of T-bet in the *Rictor*/*Foxo1* cDKO mice. Gated on three subsets of NK cells from the spleen, we found deletion of *Foxo1* in the Rictor-deficient NK cells rescued the expression of T-bet in the CD27 SP and DP subsets to the level of the corresponding WT control (**Figure 7—figure supplement 1E**). The rescue was incomplete in the terminal CD11b SP NK compartments (**Figure 7—figure supplement 1E**). Deletion of *Foxo1* alone resulted in almost a fold increasing of the T-bet protein level in all three developmental stages compared to the WT control. These data indicated that, in addition to FoxO1, other factors downstream of mTORC2 are involved in the regulation of T-bet expression.

Next, we wanted to assess whether the transcriptome-defined maturation defects in Rictor-deficient NK cells were also rescued via deletion of *Foxo1*. We profiled the CD3ε−CD122$^+$ cells from BM of littermate *Rictor*$^{fl/fl}$*Foxo1*$^{fl/+}$*Ncr1*$^{Cre/WT}$ and *Rictor*$^{fl/fl}$*Foxo1*$^{fl/fl}$*Ncr1*$^{Cre/WT}$ mice using scRNA-seq and classified the NK cells from these mice with machine-learning algorithms (**Figure 7C**). Compared to the *Rictor* cKO mouse, the *Rictor*/*Foxo1* cDKO mouse harbored reduced percentage of iNK cluster and substantially increased percentage of termNK population indicating successful rescue of the Rictor-deficient NK cells via deletion of *Foxo1* (**Figure 7C**). We further explored the expression of the immature NK genes in both samples and found reduced expression of iNK signature genes in the *Rictor*/*Foxo1* cDKO mouse compared to the *Rictor* cKO mouse (**Figure 7D**). Consistent with the rescue, we also found increased expression of termNK signature genes in the *Rictor*/*Foxo1* cDKO mouse compared to the *Rictor* cKO mouse (**Figure 7D**). The correction of the abnormal expression of iNK signature genes in the Rictor-deficient NK cells via deletion of *Foxo1* were further manifested with the reduced expression of a few mostly highly expressed iNK genes in the *Rictor*/*Foxo1* cDKO mouse sample (**Figure 7E**). Whether the restored expression of T-bet, complete loss of FoxO1, or potentially both are responsible for the correction of the iNK gene expression requires further dissection in the future.

## Discussion

The transcriptional programs that regulate NK cell development is not yet fully understood. Specifically, the transcriptional activation or repression during NK cell ontology is poorly defined. Here, we used scRNA-seq technology and found five distinct NK clusters in the BM of WT mice. Based on the expression of known markers and transcriptional similarity, we delineated the relative maturity of these five NK clusters. Based on the transcriptome information of the WT clusters, we utilized the machine-learning classifiers to define the identity of Raptor- or Rictor-deficient NK cells and found maturation profiles of these mutant NK cells distinct from the cell surface markers-based definition that we previously reported (*Yang et al., 2018*). The striking immaturity of the Rictor-deficient NK cells promoted us to evaluate the maturation of the T-bet-deficient NK cells with single-cell transcriptome analysis. We found a higher percentage of immature NK cells in *Tbx21* KO mice than the *Rictor* cKO mice and significant up-regulation of immature NK transcriptional signature in T-bet-deficient NK cells, a phenomenon that was not previously appreciated. Lastly, through deletion of *Foxo1*, we completely rescued the maturation defects of the Rictor-deficient NK cells defined by both cell surface markers and the transcriptome.

With the profiling of CD3ε−CD122$^+$ cells from the BM of WT mice, we found two distinct cell subsets, the ILC1 and *Cd3*$^{high}$ cells, besides the conventional NK population. These two cell populations presumably contribute to the heterogeneity of CD3ε−CD122$^+$NK1.1$^-$ cells, the original definition of the NKP subset. We believe the *Cd3*$^{high}$ cells are related to the NK-T cell lineage as this population contains high transcripts level of Cd3d/e/g and is nearly absent in the *Tbx21* KO mice which have fewer NK-T cells (*Townsend et al., 2004*). Further investigation of these unique *Cd3*$^{high}$ cells is warranted to determine their developmental and functional relevance. Specific to the five clusters formed by the conventional NK cells, besides the reliably identifiable least mature (iNK) and most mature (termNK) clusters, we named the rest three clusters as the transitional NK subsets. TransNK1 are transcriptionally closer to the iNK cluster and demonstrates a proliferative phenotype with high ribosomal contents and active metabolic profile. TransNK3 represents a unique NK subset with high expression of IEGs similar to the novel NK_3 cluster defined previously in the mouse spleen (*Crinier et al., 2018*). This group may be upregulating IEG's as a part of the developmental process or they may be a unique NK subset that branches off late in the developmental process.

Although we could not find reliable cell surface markers to faithfully define these five WT NK clusters, it provides an important dataset that we can utilize to profile the NK cells from mutant mice whose transcriptomic profiles are distinct from WT cells. Previously, we have reported the differential role mTORC1 and mTORC2 in regulating the development of NK cells (*Yang et al., 2018*). Based on the cell surface markers expression, we found mTORC1 is important for the early NK cell maturation from the CD27 SP to the DP stage, while mTORC2 governs the terminal maturation program, transiting from the DP to the CD11 SP stage. However, through utilizing machine-learning classifiers, we found more immature NK cells in the *Rictor* cKO mice as compared to the *Rptor* cKO mice, contradictory to the cell-surface markers-defined developmental maturity. We further profiled the T-bet-deficient NK cells as we previously proposed that the impaired expression of T-bet is responsible for the terminal maturation defect of Rictor-deficient NK cells. Strikingly, we found even more immature NK cells in the *Tbx21* KO mice and a significant up-regulation of immature NK signature genes in the T-bet-deficient NK cells, which reveals a previously unappreciated role of T-bet in suppressing the immature program of NK cells.

The further exploration of the mechanisms underlining the suppressive role of T-bet in regulating the expression of immature NK genes indicated that T-bet binds the regulatory elements of only a few immature NK genes and suppresses their transcription in mature NK cells. The increased expression of large number of immature NK genes implied additional transcription factors downstream of T-bet may play a role. We proposed that, in addition to the well-established role of FoxO1 in suppressing the expression of T-bet (*Deng et al., 2015*), T-bet can also suppress the expression of FoxO1 as the expression and activity of FoxO1 are increased in T-bet-deficient NK cells, and that this negative feedback loop is part of what transcriptionally defines mature NK cells as a lineage. Based on this information, we hypothesized that hyperactive FoxO1 in both the Rictor- and T-bet-deficient NK cells drive the expression of immature NK genes. One evidence to support this hypothesis is that deletion of Foxo1 in Rictor-deficient NK cells not only rescues the terminal maturation defects but also corrects the abnormal expression of immature NK genes. Additional experiments are required to fully establish the reciprocal regulation between FoxO1 and T-bet during the development of NK cells and their precise role in regulating the immature and terminal maturation programs.

In conclusion, we utilized scRNA-seq technology to define the developmental heterogeneity of NK cells in the BM. We further utilized these transcriptome-defined developmental clusters to reveal a previously unappreciated role of mTORC2-Akt$^{S473}$-FoxO1-T-bet axis in regulating the expression of immature NK genes during NK cell development. More importantly, through using machine-learning algorithms, we present a novel approach to define cellular differentiation.

# Materials and methods

## Mice and reagents

*Rptor*$^{fl/fl}$, *Rictor*$^{fl/fl}$, *Foxo1*$^{fl/fl}$, and *Tbx21*$^{-/-}$ mice were purchased from the Jackson Laboratory (Bar Harbor, ME). *Ncr1*$^{iCre}$ mice were a generous gift from Dr. Eric Vivier (*Narni-Mancinelli et al., 2011*). *Rptor*$^{fl/fl}$, *Rictor*$^{fl/fl}$, *Tbx21*$^{-/-}$, and *Ncr1*$^{iCre}$ mice are in C57BL/6 background. The *Foxo1*$^{fl/fl}$ mice were in FVB background. All mice were maintained in pathogen-free conditions in the Biological Resource Center at the Medical College of Wisconsin. All animal protocols were approved by Institutional Animal Care and Use Committees. The following antibodies and reagents were used in this study: CD3 (17A2), NK1.1 (PK136), CD27 (LG.7F9), CD11b (M1/70), KLRG1 (2F1), NCR1 (29A1.4), CD122 (5H4), T-bet (4B10) are from Thermo-Fisher Scientific (Waltham, MA).

## Cell separation, flow cytometry and cell sorting. Bone marrow cells were flushed, and single-cell suspensions were made by passing through the syringe/needles

Cells from spleen were prepared by gently grinding the dissected organs with micro slides (VWR, Radnor, PA). Flow cytometry analyses were conducted in LSR-II (BD Biosciences, San Jose, CA) or MACSQuant Analyzer 10 (Miltenyi Biotec, Bergisch Gladbach, Germany) and analyzed with FlowJo software (FlowJo LLC, Ashland, OR). For cell sorting, the bulk BM cells were used to isolate the CD3ε−CD122$^+$ cells for scRNA-seq experiments. Specific to the bulk RNA-seq experiment, NK cells

were first enriched using negative selection kit (STEMCELL Technologies, Vancouver, Canada). The CD27$^+$ NK cells were further sorted by FACSAria (BD Biosciences, San Jose, CA), and the purity was generally above 95%.

## Mixed bone marrow chimera

BM from CD45.1$^+$ WT mice and CD45.2$^+$ $Tbx21^{-/-}$ mice were mixed at 1:2 ratio. 10 million cell mixture was injected into each sub-lethally irradiated CD45.2$^+$$Rag1^{-/-}Il2rg^{-/-}$ recipient mouse. A pair of donor cells were injected into three recipients and the other pair of donor cells were injected into two recipients. 8 weeks later, BM CD27$^+$ NK cells from recipients were sorted and subject to bulkRNA-seq analysis (cells derived from the same donor were pooled.). Splenocytes were also collected for phenotypical analysis using flow cytometry.

## Single-cell RNA-sequencing

After sorting, cells were washed once with ice-cold PBS containing 10% FBS post-sorting and counted using hemocytometer. After that, the cells were loaded to 10X Chromium system (10X Genomics, San Francisco, CA) and run through the library preparation following guidance from the Chromium Single Cell 3' Reagent Kits v2. The libraries were quantified using NEBNext Library Quant Kit (NEW ENGLAND Biolabs, Ipswich, MA) and sequenced via Illumina NextSeq 550 (Illumina, San Diego, CA).

## scRNA-seq data analyses

After the sequencing, the raw data from each sample was demultiplexed, aligned to mm10 reference genome, and UMI counts were quantified using the 10X Genomics Cell Ranger pipeline (v2.1.1, 10X Genomics). Then, we continued the data analysis with the filtered barcode matrix files using the Seurat package (v2.3.1) (*Butler et al., 2018*) in R (v3.4.3 or above). For initial quality control, we filtered out the cells that expressed less than 200 genes or more than 2500 genes. We also removed cells with more than 5% mitochondrial transcripts content. Gene expression values for each cell were log-normalized and scaled by a factor of 10,000. To prevent clusters from being biased by cellular library size and mitochondrial transcript content, gene expression values were scaled based on the number of UMIs in each cell and the cell mitochondrial transcript content. We combined cells from all three WT mice to increase the power of unsupervised clustering analysis (*Andrews and Hemberg, 2018*). Based on the PCElbowPlot, we picked certain number of principal components (PCs) for the clustering analysis when that number reached to the baseline of the standard deviation of PC. Specific to the choice of cluster resolution, we used the 'Clustree' function to visualize the clustering progression and picked the highest resolution that still gave stable clusters (*Zappia and Oshlack, 2018*). Cell clusters were visualized using t-distributed Stochastic Neighbor Embedding plots (t-SNE). For differential gene expression, we used Model-based Analysis of Single-cell Transcriptomics (MAST) test (*Finak et al., 2015*) with genes detected in a minimum of 10% of all cells, a minimum of 0.25 average log fold-change (logFC), and a minimum of 0.05 adjusted p value. For gene set enrichment analysis (GSEA), we used fgsea function with gene sets from the Broad Institute's molecular signatures database (MSigDB) (*Subramanian et al., 2005*; *Liberzon et al., 2011*). All the p-value shown in the figures were adjusted for multiple gene sets enrichment comparison. In order to predict cellular differentiation, cells were ordered in pseudotime using Monocle2 (v2.6.4) (*Qiu et al., 2017*). All significant DEGs across five NK clusters with more than 0.25 average logFC were used to order the cells. Finally, to predict the developmental cluster of the cells from the knock out mice, we utilized one of several classifiers including: a generalized linear model, a gradient boosting model, a deep learning neural network model, and a extreme gradient boosting model, from the H2O Python Module (*Aiello et al., 2018*). Further, we used the Seurat wrapper for the random forest classifier from the ranger package (*Schwarz et al., 2010*). These classifiers were trained on a random sample made up of 80 percent of the WT NK cell non-scaled transcription data, using cross-validation and accuracy as a target end-point for the classifier to avoid overfitting. The accuracy and error of the machine-learning classifiers was measured by using the classifier to predict the developmental cluster of the remaining 20 percent of the WT NK cells which had a known cluster identity and had not been used in the initial training.

## Bulk RNA-sequencing and data analysis

The bulk RNA-seq experiment using the CD27$^+$ NK cells from *Tbx21*$^{-/-}$ and the WT mice (n=4 per group) was performed as previously described (*Yang et al., 2018*). The final libraries were sequenced via Illumina NextSeq 550 (Illumina, San Diego, CA). The dataset from the *Rictor* cKO mice was generated in the previous study (*Yang et al., 2018*). Fastq sequence data were pseudo-aligned and quantified using Salmon v0.12 (*Patro et al., 2017*), with an index built on the mm10 reference transcriptome (*Casper et al., 2018*) downloaded from the UCSC genome browser. Following pseudo-alignment, we used tximport (*Soneson et al., 2015*) and DESeq2 (*Love et al., 2014*) to analyze differential gene expression between WT and KO NK cells. Finally, we used gene set enrichment analysis via fgsea (explained above) to identify potential downstream transcription factor targets and enrichment of T-bet targets from the T-bet ChIP-Seq analysis.

## ChIP-seq analysis

The T-bet ChIP-seq dataset was published previously (*Shih et al., 2016*). After downloading the fastq files from Sequence Read Archive (SRA), the raw fastq sequence data from the T-bet ChIP sample and the input sample were aligned to the mouse genome mm10 individually using bowtie2 (*Langmead and Salzberg, 2012*). Significant T-bet binding peaks were identified using MACS2 with a peak q-value cutoff of $1 \times 10^5$ (*Zhang et al., 2008*). These significant peaks were then annotated using the 'annotate peaks' tool from the HOMER package (*Heinz et al., 2010*). A gene set of potential T-bet target genes was produced by selecting protein-coding genes with significant T-bet peaks within the gene-body or promoter. This gene set of potential T-bet targets was used in GSEA analyses with both combined WT scRNAs-seq dataset and the CD27/CD11b bulk RNA-seq dataset.

## Acknowledgements

We thank Lucia Sammarco and her Lulu's Lemonade Stand for inspiration, motivation and support. This work was supported in part by NIH R01 AI102893 and NCI R01 CA179363 (SM and MST); HRHM Program of MACC Fund (SM and MST), Nicholas Family Foundation (SM); Gardetto Family (SM); and MACC Fund (MST and SM). *Ncr1*$^{iCre}$ mice were a kind gift from Dr Eric Vivier, Centre d'Immunologie de Marseille-Luminy, France. CY is the inaugural recipient of the J Evan Sadler Graduate Scholar Award in the Blood Research Institute and we extend our special thanks to Dr. Sadler's family.

## Additional information

### Funding

| Funder | Grant reference number | Author |
| --- | --- | --- |
| National Institutes of Health | R01 AI102893 | Subramaniam Malarkannan<br>Monica S Thakar |
| National Cancer Institute | R01 CA179363 | Subramaniam Malarkannan<br>Monica S Thakar |
| MACC Fund | HRHM Program | Subramaniam Malarkannan<br>Monica S Thakar |
| Nicholas Family Foundation | | Subramaniam Malarkannan |
| Gardetto Family | | Subramaniam Malarkannan |
| MACC Fund | | Monica S Thakar<br>Subramaniam Malarkannan |
| Blood Research Institute | Graduate Scholar Award | Chao Yang |

The funders had no role in study design, data collection and interpretation, or the decision to submit the work for publication.

## Author contributions
Chao Yang, Conceptualization, Data curation, Formal analysis, Validation, Investigation, Visualization, Methodology; Jason R Siebert, Conceptualization, Data curation, Software, Formal analysis, Validation, Investigation, Visualization, Methodology; Robert Burns, Software, Formal analysis, Visualization, Methodology; Yongwei Zheng, Ao Mei, Benedetta Bonacci, Data curation; Demin Wang, Supervision; Raul A Urrutia, Matthew J Riese, Sridhar Rao, Karen-Sue Carlson, Conceptualization, Resources, Investigation; Monica S Thakar, Conceptualization, Resources, Supervision, Funding acquisition, Investigation; Subramaniam Malarkannan, Conceptualization, Resources, Supervision, Funding acquisition, Validation, Investigation, Visualization, Methodology, Project administration

## Author ORCIDs
Subramaniam Malarkannan (iD) https://orcid.org/0000-0002-7511-2731

## Ethics
Animal experimentation: All mice were maintained in pathogen-free conditions in the Biological Resource Center at the Medical College of Wisconsin. All animal protocols were approved by Institutional Animal Care and Use Committees. The unique animal protocols that are approved by the IACUC and used in this study is: AUA1512.

## Decision letter and Author response
Decision letter https://doi.org/10.7554/eLife.51339.sa1
Author response https://doi.org/10.7554/eLife.51339.sa2

# Additional files
## Supplementary files
• Supplementary file 1. DEGs of five WT NK clusters. Related to *Figures 1* and *2*.

• Supplementary file 2. DEGs of clusters formed by WT and Raptor-deficient cells. Related to *Figure 3*.

• Supplementary file 3. DEGs of clusters formed by WT and Rictor-deficient cells. Related to *Figure 3*.

• Supplementary file 4. DEGs of clusters formed by WT and T-bet-deficient cells. Related to *Figure 5*.

• Transparent reporting form

## Data availability
Sequencing data have been deposited in GEO under accession code GSE150166.

The following dataset was generated:

| Author(s) | Year | Dataset title | Dataset URL | Database and Identifier |
|---|---|---|---|---|
| Yang C, Siebert JR, Burns R, Zheng Y, Mei A, Bonacci B, Wang D, Urrutia RA, Riese MJ, Rao S, Carlson K, Thakar MS, Malarkannan S | 2020 | Single-cell transcriptome reveals the novel role of T-bet in suppressing the immature NK gene signature the immature NK gene signature | https://www.ncbi.nlm.nih.gov/geo/query/acc.cgi?acc=GSE150166 | NCBI Gene Expression Omnibus, GSE150166 |

The following previously published datasets were used:

| Author(s) | Year | Dataset title | Dataset URL | Database and Identifier |
|---|---|---|---|---|
| Yang C, Tsaih SW, | 2018 | mTORC1 and mTORC2 | https://www.ncbi.nlm. | NCBI BioProject, |

| Lemke A, Flister MJ, Thakar MS, Malarkannan S | | differentially regulate NK cell development | nih.gov/bioproject/?term=PRJNA434424 | PRJNA434424 |
| Shih HY, Sciume G, Mikami Y, Guo L, Sun HW, Brooks SR, Urban JF, Davis FP, Kanno Y, O'Shea JJ | 2016 | Developmental Acquisition of Regulomes Underlies Innate Lymphoid Cell Functionality | https://www.ncbi.nlm.nih.gov/geo/query/acc.cgi?acc=GSE77695 | NCBI Gene Expression Omnibus, GSE77695 |

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
