## [Decision Letter]

**Acceptance summary:**

The editors and the reviewers recognised the novelty and the importance of your work describing new insights in NK development and the transcriptional program regulating T-bet in NK cell differentiation. Your data represent an interesting resource for the community to investigate and mine.

**Decision letter after peer review:**

Thank you for submitting your article "Single-cell transcriptome reveals the novel role of T-bet in suppressing the immature NK gene signature" for consideration by *eLife*. Your article has been reviewed by Satyajit Rath as the Senior Editor, a Reviewing Editor, and two reviewers. The following individuals involved in review of your submission have agreed to reveal their identity: Nicholas Huntington (Reviewer #1).

The reviewers have discussed the reviews with one another and the Reviewing Editor has drafted this decision to help you prepare a revised submission.

Summary:

This manuscript provides new insights in NK development and the transcriptional program regulating T-bet in NK cell differentiation and represent an interesting resource for the community to investigate and mine. Aiming to understand better the role of T-bet in defining NK cell maturation and using single cell RNA-seq and bulk population RNA-seq of NK cells from *Tbx21-/-* mice, the authors identify concordance in expression between Rictor KO and Tbet KO NK cells. This allowed the identification of a novel role of mTorc2/FoxO1/Tbet axis in the maturation program of NK cells.

Essential revisions:

Below are the essential revision requirements to address:

1) We recommend the use of *Tbx21* conditional mice to validate the essential observations made as complete knockout mice have been used for the T-bet-deficient analyses and indirect effect due the full KO context may have additional phenotypes that could impact on the data. For example, a recent paper in Cell described expression of Tbx21 in conventional type 2 dendritic cells. Alternatively mixed bone marrow chimeras (WT:Tbx21-/-) would resolve this issue.

2) How were the Bcl2 and BH3 only member impacted by loss of Rictor? Is Foxo1 rescuing this phenotype via regulation of Bcl2l11? Foxo1 has been shown to regulate Bcl2l11 in NK cells.

3) Only one mouse per condition was sequenced for the single-cell analysis. This should be consolidated. The authors should explain how they correct for batch effect as well as explain why a LogFC > 0.25 was used as a cutoff for differential gene expression as Typically 1.5 or 1.2 is used. They should also provide the average number of genes for samples.

[Editors' note: further revisions were suggested prior to acceptance, as described below.]

Thank you for sending your article entitled "Single-cell transcriptome reveals the novel role of T-bet in suppressing the immature NK gene signature" for peer review at *eLife*. Your article is being evaluated by peer reviewers, and the evaluation is being overseen by a Reviewing Editor and Satyajit Rath as the Senior Editor.

The consensus reached between us suggests that to reach a favourable decision, data from either the use of Tbx21-conditional mice, or mixed Tbx21 bone marrow chimeras, are needed to avoid any possible indirect and unforeseen effects.

Given the need for this essential revision, the editors and reviewers invite you to respond with an action plan and timetable for the completion of the additional work. We plan to share your responses with the reviewers and then issue a formal decision. Do please note that *eLife* understands the current difficult situation, so please do not hesitate to let us know realistic estimates.

---

## [Author Response]

Summary:This manuscript provides new insights in NK development and the transcriptional program regulating T-bet in NK cell differentiation and represent an interesting resource for the community to investigate and mine. Aiming to understand better the role of T-bet in defining NK cell maturation and using single cell RNA-seq and bulk population RNA-seq of NK cells from Tbx21-/- mice, the authors identify concordance in expression between Rictor KO and Tbet KO NK cells. This allowed the identification of a novel role of mTorc2/FoxO1/Tbet axis in the maturation program of NK cells.Essential revisions:Below are the essential revision requirements to address:1) We recommend the use of Tbx21 conditional mice to validate the essential observations made as complete knockout mice have been used for the T-bet-deficient analyses and indirect effect due the full KO context may have additional phenotypes that could impact on the data. For example, a recent paper in Cell described expression of Tbx21 in conventional type 2 dendritic cells. Alternatively mixed bone marrow chimeras (WT:Tbx21-/-) would resolve this issue.

We appreciate the Reviewer’s suggestion. We utilized the global T-bet-deficient mice as this was commercially available to us. Our requests to obtain the conditional knockout mice are in the process and is taking more time than we expected. Meanwhile, we have obtained additional global knockout mice towards performing bone marrow chimeras as suggested by the Reviewer. Irrespective of this, we worry that: a) this delay may negatively affect the ability to publish our work in a timely manner and b) our work on this direction may only be validation and may not result in a different set of findings.

We request the Editor and the Reviewer to kindly assess our request. Thank you.

2) How were the Bcl2 and BH3 only member impacted by loss of Rictor? Is Foxo1 rescuing this phenotype via regulation of Bcl2l11? Foxo1 has been shown to regulate Bcl2l11 in NK cells.

We thank review’s comment. The expression of BH3 only member Bim (encoded by *Bcl2l11*) is increased in Rictor-deficient NK cells which may results from hyperactive FoxO1 (see Author response image 1). However, the expression of anti-apoptotic protein Bcl2 is also proportionally increased in Rictor-deficient NK cells (see Author response image 1). We did not observe increased cell death in Rictor-deficient NK cells (Figure 2—figure supplement 1B in Yang, 2018). In addition, we did not observe changes in the mRNA level of other BH3 only member proteins (Bid/Bad) or other pro-apoptotic regulators (Bax/Bak1) in Rictor-deficient NK cells (see Author response image 1). Therefore, we conclude FoxO1 did not rescue this phenotype via regulation of Bim (encoded by *Bcl2l11*).

3) Only one mouse per condition was sequenced for the single-cell analysis. This should be consolidated. The authors should explain how they correct for batch effect as well as explain why a LogFC > 0.25 was used as a cutoff for differential gene expression as Typically 1.5 or 1.2 is used. They should also provide the average number of genes for samples.

We thank reviewer’s comment. When we performed single-cell RNA-seq experiments, we only used the WT and KO littermates which reduces the biological variation. About 2,000 cells from each genotype were sequenced which ensures the full representation of the population. More importantly, we have bulk RNA-seq analyses of Rictor- or T-bet-deficient NK cells with at least three biological replicates that strongly support our single-cell analysis.

Regarding the batch effect, there is no batch effect in comparison of each individual *Rptor*, *Rictor* or *Tbx21* KO NK cells with the corresponding WT NK cells as the WT and KO is always done in the same experiments. When we analyze the WT NK cells only (Figure 1 and Figure 2), we combined cells from all three WT mice to increase the power of unsupervised clustering analysis. We scaled the sample variance among those three WT samples which in the same time scaled the batch variation.

The default LogFC threshold of scRNA-seq analysis in the Seurat package is 0.25. The typical 1.5 to 1.2 cut-off is used in bulk RNA-seq analysis.

In Figure 1—figure supplement 1C, we have showed the nUMI, nGene and percent.mito of all samples.

[Editors' note: further revisions were suggested prior to acceptance, as described below.]

The consensus reached between us suggests that to reach a favourable decision, data from either the use of Tbx21-conditional mice, or mixed Tbx21 bone marrow chimeras, are needed to avoid any possible indirect and unforeseen effects.

We have conducted mixed bone marrow chimera experiment using BM cells from CD45.1^+^ WT mice and CD45.2^+^*Tbx21^-/-^* mice with sublethally irradiated CD45.2^+^*Rag1^-/-^Il2rg^-/-^* as recipient mice. Eight weeks later, BM CD27^+^ NK cells from recipients were sorted and subject to bulk RNA-seq analysis (cells derived from the same donor were pooled.). Splenocytes were also collected for phenotypical analysis using flow cytometry. As shown in Figure 5—figure supplement 2A and 2B, we have confirmed the terminal maturation defects indicated by the loss of CD11b SP population and diminished expression of KLRG1 on NK cells derived from *Tbx21^-/-^* donors, consistent with previous report (Townsend, 2004).

The analysis of bulk RNA-seq data revealed up-regulation of immature NK signature genes and reduced expression of terminally mature NK signature genes as shown in heatmap from Figure 5-figure supplement 2C. The expression pattern is similar to what we observed in bulk RNA-seq data derived from global Tbx21 KO mice (Figure 5F). This data indicates that Tbet intrinsically suppresses the expression of immature NK genes.